# Photosynthetic light requirement near the theoretical minimum detected in Arctic microalgae

Clara J. M. Hoppe [1] ✉, Niels Fuchs [2], Dirk Notz [2], Philip Anderson [3], Philipp Assmy[4], Jørgen Berge [5], Gunnar Bratbak [6], Gaël Guillou[7], Alexandra Kraberg[1], Aud Larsen [8], Benoit Lebreton[7], Eva Leu[9], Magnus Lucassen [1], Oliver Müller [6], Laurent Oziel [1], Björn Rost [1,10], Bernhard Schartmüller [5], Anders Torstensson [11,12] & Jonas Wloka[13]

Photosynthesis is one of the most important biological processes on Earth, providing the main source of bioavailable energy, carbon, and oxygen via the use of sunlight. Despite this importance, the minimum light level sustaining photosynthesis and net growth of primary producers in the global ocean is still unknown. Here, we present measurements from the MOSAiC field campaign in the central Arctic Ocean that reveal the resumption of photosynthetic growth and algal biomass buildup under the ice pack at a daily average irradiance of not more than $0.04 \pm 0.02\,\mu mol$ photons $m^{-2}\,s^{-1}$ in late March. This is at least one order of magnitude lower than previous estimates ($0.3$–$5\,\mu mol$ photons $m^{-2}\,s^{-1}$) and near the theoretical minimum light requirement of photosynthesis ($0.01\,\mu mol$ photons $m^{-2}\,s^{-1}$). Our findings are based on measurements of the temporal development of the under-ice light field and concurrent measurements of both chlorophyll $a$ concentrations and potential net primary production underneath the sea ice at 86 °N. Such low light requirements suggest that euphotic zones where photosynthesis can occur in the world's oceans may extend further in depth and time, with major implications for global productivity estimates.

The distribution of life on Earth is critically linked to environmental conditions suitable for photosynthesis. In the ocean, photosynthesis can only occur in the uppermost euphotic layer where sufficient light is available. Therefore, the extent of the euphotic zone, where photosynthesis exceeds respiration and other loss terms[1], is a key control for oceanic net primary production (NPP) and major biogeochemical cycles[2]. Despite this importance, the minimum light requirement for photosynthetic growth, and the according maximum depth of the euphotic zone, is still not well constrained. In this study, we address this gap by presenting complementary field measurements of under-ice irradiances and phototrophic biomass from the high Arctic that show NPP to be possible at extremely low light levels.

[1]Alfred-Wegener-Institut—Helmholtz-Zentrum für Polar- und Meeresforschung, Bremerhaven, Germany. [2]Center for Earth System Sustainability, Institute of Oceanography, University of Hamburg, Hamburg, Germany. [3]Scottish Association for Marine Science, Oban, Scotland. [4]Norwegian Polar Institute, Fram Centre, Tromsø, Norway. [5]UiT The Arctic University of Norway, Tromsø, Norway. [6]University of Bergen, Bergen, Norway. [7]Joint Research Unit Littoral, Environment and Societies (CNRS—University of La Rochelle), La Rochelle, France. [8]NORCE Norwegian Research Centre, Bergen, Norway. [9]Akvaplan-niva, Tromsø, Norway. [10]Faculty of Biology/Chemistry, University Bremen, Bremen, Germany. [11]Department of Aquatic Sciences and Assessment, Swedish University of Agricultural Sciences, Uppsala, Sweden. [12]Swedish Meteorological and Hydrological Institute, Community Planning Services—Oceanography, Västra Frölunda, Sweden. [13]Independent Researcher, Bremen, Germany. ✉e-mail: Clara.Hoppe@awi.de

Traditionally, the lower limit of the euphotic zone is defined as the depth at which light levels are 1% of their surface value[1], which, in absolute terms, can be as high as 20 μmol photons m$^{-2}$ s$^{-1}$ for midday conditions at the equator with a typical surface value of 2000 μmol photons m$^{-2}$ s$^{-1}$. However, it has long been known that this threshold overestimates the minimum light requirement for NPP[3], implying that the euphotic zone actually extends to greater depths. For the absolute lower light requirement for photosynthetic growth, theoretical considerations suggest a light level of around 0.01 μmol photons m$^{-2}$ s$^{-1}$[4–6]. Below this threshold, photosynthetic carbon fixation is considered impossible due to biochemical constraints[6]. However, it is unclear how far from this threshold photosynthesis indeed occurs in nature. In the ocean, NPP has been observed at light levels as low as 0.3–5 μmol photons m$^{-2}$ s$^{-1}$[5,7–10], i.e., 30–500 times higher than the theoretical minimum threshold. For ice algae, thresholds of 0.17 and 0.36 μmol photons m$^{-2}$ s$^{-1}$ have been reported[4,11], still exceeding the theoretically derived minimum threshold by far. While differences between these measurements may seem small in absolute terms, the light levels of large fractions of the world's oceans are in these ranges, at least for some part of the year. Therefore, knowledge of the lower light limit at which NPP can occur is a prerequisite to realistically estimate the temporal and spatial distribution of primary production and concurrent ecosystem dynamics.

Here, we address this knowledge gap by analyzing field measurements from the Multidisciplinary drifting Observatory for the Study of the Arctic Climate (MOSAiC), during which an interdisciplinary team of researchers on RV Polarstern drifted across the Arctic Ocean with the sea ice for one year (September 2019–October 2020)[12,13]). Such a year-round survey allowed us to study the resumption of photosynthetic activity after the hiatus of the polar night, as sunlight returned to the central Arctic Ocean (at 88–84°N) during the period February to April 2020 (Fig. S1). For our study, data were collected at different locations on the pack ice around the ship, while the ice drifted over the mixed layer of the ocean (Figs. 1, S2). We used three different methods to determine the potential onset of photosynthetic activity during that period: measurements of potential primary production, determination of chlorophyll $a$ (Chl-a) concentrations, and direct cell counts of photosynthetic microalgae. In the following, we discuss the results of these measurements as a function of time before linking them to the related light levels.

## Results and Discussion

The most sensitive method we employed to determine the potential onset of photosynthesis underneath the ice is measuring the potential $^{14}$C productivity (termed NPP, in the following) of the ambient phytoplankton assemblages for 24 h using $^{14}$C-based volume-specific carbon fixation rates. Surface water samples were exposed to reference conditions of a temperature of 1 °C and an irradiance of 10 μmol photons m$^{-2}$ s$^{-1}$ in the laboratory. NPP rates significantly above dark controls were measured as early as in January and February, indicating physiologically active overwintering assemblages that can resume photosynthesis as soon as they are exposed to light (Fig. S3). Surprisingly, the onset of an exponential increase of potential NPP occurred in the mixed layer before March 14th (Fig. 2A). The lack of trends in carbon fixation to Chl-a (Fig. S4, Mann–Kendall Trend test) and particulate organic carbon (POC) to Chl-a ratios (Fig. S5, Mann–Kendall Trend test) throughout the study period indicate that this signal does not relate to physiological acclimation. This is in line with field observations[7,14,15] and laboratory studies[16–18], which found cells to retain their photosynthetic machinery active over the polar night so that it can be instantly used upon the return of light. Even though our measurements represent only the potential capacity of the existing biomass to fix carbon rather than the actual in situ rates, the increase in potential NPP that we identified from March 14th onwards is highly indicative of an actual increase in photosynthetic biomass despite the extremely low light levels this early in the season.

As a more direct measure of in situ biomass production through photosynthesis, we used surface ocean Chl-a concentrations from discrete underway samples. These were obtained from 11 m water depth on a daily basis, which allowed us to derive a highly resolved time series of Chl-a-based biomass in the upper mixed layer of the ocean (Fig. 2B). This conversion of Chl-a concentration into biomass is possible because Chl-a specific carbon fixation and the ratio of POC to Chl-a did not display a trend over the study period (Figs. S4, S5, Mann–Kendall Trend test).

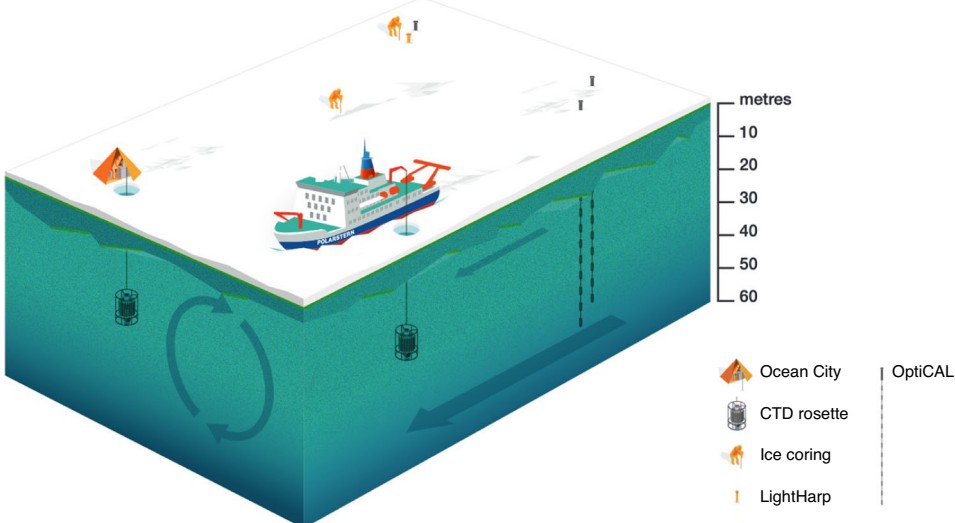

metres
— 10
— 20
— 30
— 40
— 50
— 60

Ocean City OptiCAL

CTD rosette

Ice coring

LightHarp

**Fig. 1 | Schematic illustration of sampling sites and depths.** MOSAiC sampling sites and depths around RV Polarstern during spring 2020 relevant to this study are illustrated in a schematic way. Biological parameters from the upper mixed layer (circular arrows) were collected from 20 m depth via rosettes deployed through holes in the ice at Ocean City and next to RV Polarstern (see refs. 13,24), as well as the ship's underway system at 11 m depth. Sea ice cores for sea ice Chlorophyll $a$ and Net Primary Production $P$ were collected at the first-year ice coring site. Light measurements were collected with OptiCALs (Optical Chain And Logger) gg, hh, and ee down to 50 m water depth as well as the lightharp, the latter inside the ice column only. Exact locations of the different sampling sites are illustrated in Fig. S2. Please note that the ice was drifting about six times faster than the underlying water column[23] so that specific locations on the pack ice do not represent different sampling locations in the ocean.

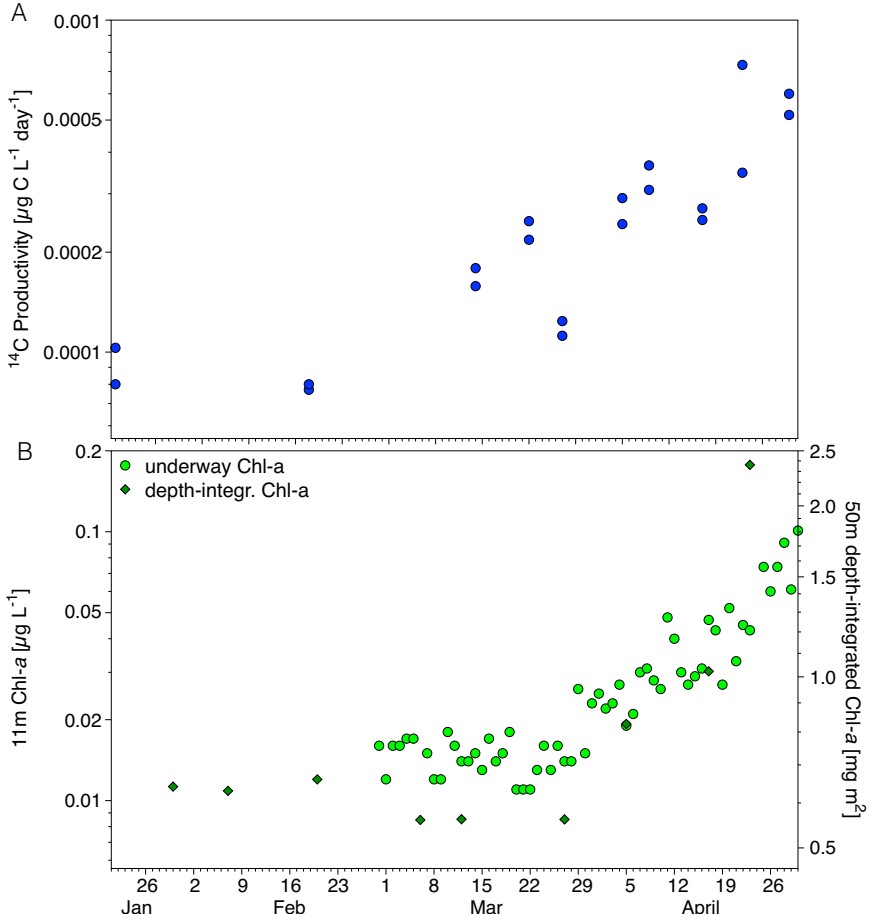

**Fig. 2 | Temporal development in phytoplankton activity and biomass.** Development of (**A**) potential net $^{14}$C productivity (as a proxy for Net Primary Production) under reference conditions, and (**B**) 11 m water column Chlorophyll $a$ (Chl-a) concentrations (light green) and 50 m depth-integrated Chl-a standing stocks (dark green) as a function of time. Chl-a-specific $^{14}$C productivity production is displayed in Fig. S4.

Chl-a concentrations measured during the first half of March were similar to those measured during February (i.e., around 0.01–0.02 µg L$^{-1}$), but then increased exponentially from mid-March to the end of April, reaching 0.05 µg L$^{-1}$ by mid-April. We applied a change-point analysis package[19] to this data to estimate the onset of photosynthetic growth. In line with a visual inspection of the time series (Fig. 2B), six independent models from this package consistently identified March 27th ± 1 day as the change point in the Chl-a time series. Opting for the more conservative estimate, we identified March 28th as the onset of photosynthetic growth in this dataset. The rate of increase after this date fits well to the rate of change in surface Chl-a concentrations calculated over 5-day intervals in our time series[20]: While the rate of change varied between slightly negative and slightly positive values during most of March, rates were positive for nine consecutive days from March 25th and remained mainly positive at higher rates for the rest of the study period (Fig. S6). Congruently, accumulation rate constants based on underway Chl-a over 7-day intervals were −0.01 ± 0.04 d$^{-1}$ before March 27th and +0.06 ± 0.07 d$^{-1}$ after the change point. Depth-integrated Chl-a from CTD-rosette samples (upper 50 m, Fig. 2B), being available only in lower temporal resolution, followed a similar trend with standing stocks increasing between March 27th and April 5th for the first time, with standing stocks of 0.6 and 0.8 mg m$^{-2}$, respectively. In summary, as a conservative estimate, March 28th marked the onset of photosynthetic growth in the water column underneath the ice cover as determined from the measurement of Chl-a concentration.

As a third method to determine the increase in primary production and the subsequent onset of photosynthetic growth, we used cell counts via light microscopy and flow cytometry (FCM) of water samples from 20 m water depth (Fig. S7). Given the need to buildup biomass before cell division, we expected to see cell counts to increase sometime after the actual onset of net photosynthesis. Consistent with this expectation, cell counts indeed increased a few weeks after the identified onset of net photosynthesis on March 28th, mainly driven by the pennate diatom *Pseudo-nitzschia* spp. (Table S1).

Additionally, we also examined the temporal development of ice algal communities[21] during the study period. These were sampled in lower temporal resolution owing to the inaccessibility of the ice coring areas under the prevailing harsh weather and ice conditions[12]. Chl-a concentrations increased ten-fold in the bottom 10 cm of first-year sea ice from mid-February to the end of March (Fig. S8). Similarly, FCM cell counts indicated a 5 to 10-fold increase in photosynthetic cell numbers between these sampling dates, mainly driven by an increase in the nanoplanktonic size class (2–20 µm; Fig. S9). Light microscopic cell counts confirmed this pattern, indicating that different pennate diatoms (mainly *Nitzschia* spp., and *Navicula* spp.) dominated this early increase in the ice algal assemblage (Table S1). Using the exponential rate of increase from the end of March to the end of April ($r^2 = 0.86$), we estimate that the first doubling of Chl-a concentrations in first-year bottom ice occurred as early as March 11th. Also, potential ice-algae NPP (Fig. S8) increased by an order of magnitude between February and late March.

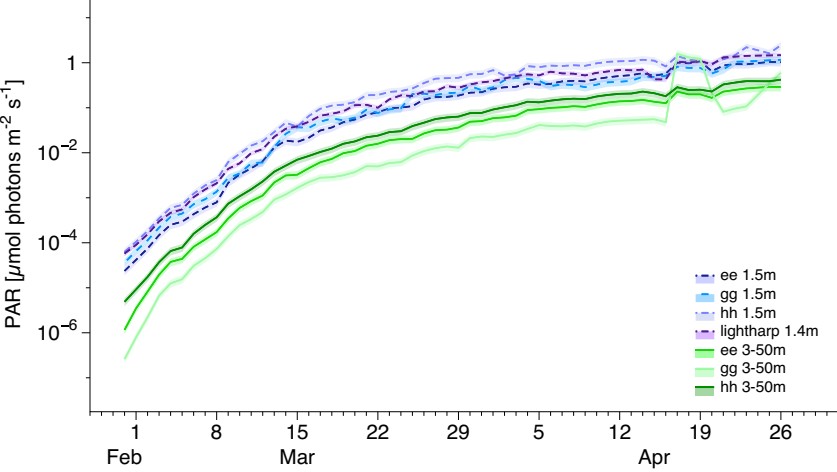

**Fig. 3 | Temporal development in light availability.** Photosynthetically active radiation (PAR) at the sea ice bottom (blue) and in the water column (3–50 m depth-integrated values; green) for the different OptiCALs (Optical Chain And Logger sensors ee, gg, hh) and light harp as a function of time. Shaded areas indicate error bands of 20% and 12.2% uncertainty for OptiCALs and lightharp, respectively. A lead that opened on April 17th near sensor gg strongly increased light availability in the water column at this location for 3–4 days.

Having established that photosynthetic growth started between mid-March and early April, depending on the sensitivity of the detection method used, we will now discuss the light levels under which this early photosynthesis took place. For this analysis, we draw on a number of optical sensors that were deployed during the MOSAiC expedition: above, within, and underneath the ice (Figs. 1, S2). These sensors provided continuous PAR measurements during the transition from polar night to polar day (Fig. 3).

To obtain the light intensity in the water column at which NPP set in, we used measurements from three OptiCAL (Optical Chain And Logger) autonomous ice-tethered observatories that sampled downwelling irradiance in the spectrally integrated photosynthetic active radiation (PAR) wavelength range 400–700 nm [$\mu$mol photons m$^{-2}$ s$^{-1}$][22]. These light chains with nominal sensor depths of 1, 2, 3, 4, 5, 6, 7, 9, 14, 21, 32, and 50 m below the ice surface were deployed at three different locations, allowing us to estimate the spatial variability of the under-ice light field. To determine the mean irradiance that primary producers in the water column were exposed to, we averaged the light fields horizontally across these three sites that continuously drifted with the ice over the underlying ocean[23,24]. Vertically, we averaged our measurements across the upper 50 m of the water column. This vertical averaging is motivated by the fact that the Chl-a profiles varied little over this depth range, indicating that cells were likely mixed homogeneously in the upper 50 m and experienced the average light level across this depth range[25,26]. In addition, this depth range agrees well with an independently estimated mixed layer depth of 40–65 m[23,27] and a Brunt-Väisälä frequency maximum between 50 and 60 m[23,27]. We consider the mean light field 24 h prior to the biological sampling, as NPP is an integrated signal commonly quantified on a daily basis. To account for potential horizontal inhomogeneities in the translucency of the ice that could have caused an area-averaged PAR above the specific measurement levels at the sensor sites, we have added generous uncertainty margins to all light measurements, thus ensuring that the values reported here provide an absolute upper bound on the true light intensity that the cells were exposed to local shading effects were accounted for by using a correction model for the euphotic zone under sea ice[28], while the impact of inhomogeneities in the snow cover was accounted for by adding 33% to all light-intensity measurements, a correction term that has been derived from the statistical distribution of snow thicknesses in the measurement area (see "Methods" for details).

Using this approach, we find that PAR values underneath first-year ice (thickness: 1.14–2.05 m[21]) were similar to those under second-year ice (thickness 1.71–2.44 m[29]). This is related to the dominating impact of the snow cover on the under-ice light field[30]. The snow cover at the MOSAiC site varied little between ice areas of different ages and barely changed between February and the onset of melt in May[31]. Spatial differences in snow depth and thus light availability were instead primarily related to the dynamic deformation of the ice cover, contradicting the common notion that first-year ice generally supports higher primary production due to the assumed larger light availability[32,33]. For the purpose of this study, we therefore do not differentiate between first and second-year ice.

The light measurements indicate that our conservative estimate of the first day of positive water column Chl-a accumulation on March 28th was associated with a daily average PAR of not more than $0.04 \pm 0.02$ $\mu$mol photons m$^{-2}$ s$^{-1}$. After this, Chl-a concentration increased exponentially with increasing PAR (Fig. 4). Given that alternative energy sources such as respiration of storage compounds or mixotrophic uptake of organic matter have been available throughout winter and are not related to the return of sun light[18,34,35], the exponential increase in Chl-a concentrations must be a direct response to the measured increase in PAR.

The conservatively estimated PAR value of 0.04 $\mu$mol photons m$^{-2}$ s$^{-1}$ is two to three orders of magnitude lower than the 0.415 mol photons m$^{-2}$ d$^{-1}$ isolume commonly used to constrain the zone of NPP in biogeochemical models[36–38], one order of magnitude lower than the lowest previous record for ice algae[4], and two orders of magnitude lower than previous estimates from surface waters[7]. This surprisingly low PAR value is very close to the theoretical minimum photon requirement for oxygenic photosynthesis of 0.01 $\mu$mol photons m$^{-2}$ s$^{-1}$ [4–6], and provides evidence that evolution has optimized the efficiency of photosynthesis in microalgae to an astonishing degree.

Our finding that NPP can occur at extremely low light levels has substantial implications for our understanding and quantification of processes driving ocean ecosystems, not only in the Arctic but potentially in all light-limited marine habitats on Earth. For the Arctic, knowing the lower light limit for microalgae is crucial for understanding the timing of primary production and matching with their zooplankton grazers and higher trophic levels[39]. Even though the measured early spring Chl-a biomass levels in the MOSAiC study region were very low, they are comparable to typical Arctic summer surface Chl-a concentrations (>80°N, $0.33 \pm 0.55$ $\mu$g L$^{-1}$; $n = 1487$[40–42]), and represent a substantial reservoir of organic material that higher

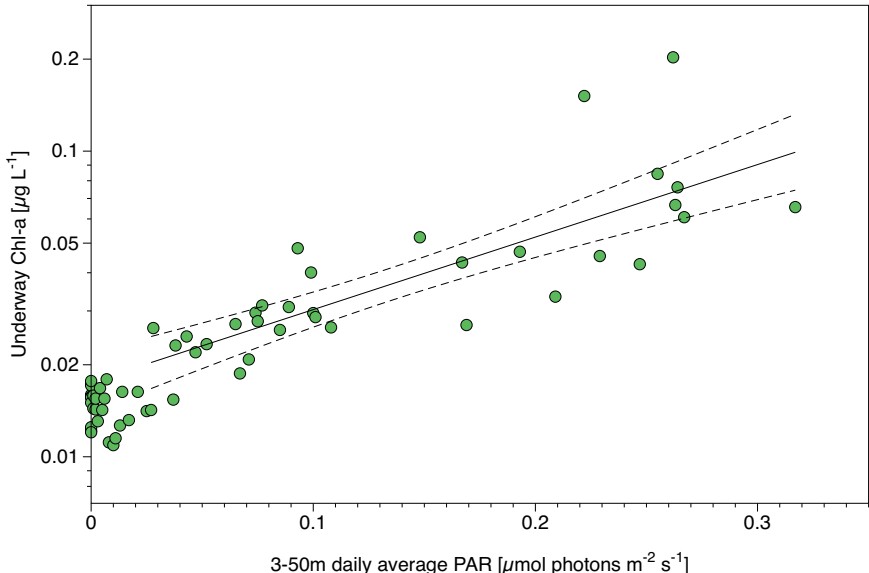

**Fig. 4 | Relationship between photosynthetic biomass and light availability.**
Underway Chlorophyll *a* (Chl-a) concentrations (11 m depth; log scale) as a function of 3–50 m depth-integrated daily average photosynthetically active radiation (PAR) values with exponential fits of the period before ($r^2 = 0.03$ indicating no relationship between Chl-a and PAR) and after ($r^2 = 0.66$ indicating an exponential increase in Chl-a with increasing PAR) the change point determined after Beaulieu & Killick[18]. A lead that opened on April 17th at sensor gg strongly increased light availability at this location for 3–4 days (see Fig. 3). The respective data points were excluded from this calculation (see Fig. S9 for full data).

trophic levels can rely on. In addition, the early production of biomass that we found in bottom ice led to Chl-a concentrations of more than 5 µg L$^{-1}$ in early April, substantially more than the average Chl-a concentration of $1.0 \pm 1.3$ µg L$^{-1}$ previously observed during summer in sea ice of the Central Arctic Ocean (>80°N, $n = 112$[40,43,44]). This implies that the high Arctic Ocean ecosystem may be similarly fueled by such low light productivity in the severely undersampled spring and fall, with the short summer period being less important than previously thought. Our early spring measurements of phototrophic biomass underneath the ice and snow cover illustrate that satellite-derived Chl-a and NPP in ice-covered waters[45] might be underestimating early season biomass buildup. Our results also challenge the common notion that summer microalgal growth in the perennially ice-covered Arctic Ocean is light-limited[45]. Instead, nutrient limitation[46] and grazing pressure[47] might be the primary controlling factors of biomass buildup during the polar day.

Our data from the Arctic Ocean indicate that net photosynthetic biomass buildup close to the theoretical minimum light requirement is physiologically possible, despite certain losses for example due to grazing. While the specific conditions of the Arctic polar night and the return of sunlight may represent a unique opportunity to quantify this light intensity threshold in situ, the fundamental capacity of photosynthesis to be this efficient likely also applies to other low light environments. Our results thus imply that phytoplankton communities in the world's oceans during low light seasons and at greater depths throughout the year may be significantly more productive than previously thought. As a consequence, the depth of the euphotic zone in large parts of the world's oceans may be considerably deeper than previously reported, making our measurements highly relevant for understanding life in the ocean's twilight zones. To quantify this possible impact, our estimate corresponds to a much larger euphotic zone depth with a threshold of 0.003% of an incoming irradiance of 2000 µmol photons m$^{-2}$ s$^{-1}$. In fact, this would translate into euphotic zone depths more than twice as deep compared to the 1% criterion. Using the previous examples and assuming, for example, an attenuation coefficient of 0.2 m$^{-1}$, our threshold would deepen the bottom of the euphotic zone from 23 to 54 m. This substantially increases the vertical extent and thus the total volume of the euphotic zone in the

world's oceans[3] and may change our view on upper twilight zone ecology and biogeochemistry[48]. As such, our finding that net photosynthesis is possible at extremely low light levels can have consequences for our understanding of the entire world's oceans and points towards large potentials for hidden and previously overseen marine primary production.

## Methods

The measurements reported in this study were conducted in the framework of the Multidisciplinary Drifting Observatory for the Study of Arctic Climate (MOSAiC) drift expedition in the Central Arctic Ocean. During the study period February to April 2020, RV Polarstern[49] drifted with the sea ice from 88°N to 84°N[12,45] (see Figs. S1 and S2).

### Water column sampling

Samples from the upper mixed layer were collected approximately once per week at 20 m depth from Niskin bottles either from the ship's CTD rosette, or a smaller rosette deployed on the ice floe[13,24]. Additional upper mixed layer samples with higher temporal resolution were taken daily from the ship's underway system with an intake at 11 m water depth.

### Sea ice sampling

Sea ice samples were collected from level first-year ice (FYI) at a site located ~1.5 km away from the ship, where light pollution from the ship was too low to be measurable even during the polar night[12,13]. Cores were collected using a Kovacs Mark II 9 cm inner diameter corer and were sectioned from both the ice surface and bottom directly in the field. To allow for co-located measurements of key parameters from the same samples, sections from 3–4 ice cores were pooled in Whirl-Pack® bags. Back on the ship, ice samples ($n = 1$ per sampling event) were further processed in a 1 °C cold container under dim light conditions (NPP samples were only handled under red light). For salinity-buffered melting[50], 500 mL of 0.2 µm filtered surface seawater (11 m) was added per 10 cm ice. NPP samples consisted of less consolidated bottom sections only, which were crushed and melted in the dark at room temperature for 1–2 h. Pooled subsamples for all other parameters were melted in the dark at room temperature overnight. The

total volume of the melted pools was measured before samples were split for different sample pipelines. Total volumes, filtered seawater, and subsample volumes were used to calculate corrected sample volumes.

## Chlorophyll *a* analysis

Samples for the analysis of Chlorophyll *a* (Chl-a) were filtered onto pre-combusted glass fiber filters (GF/F; 0.7 μm nominal pore size) using mild vacuum. Samples ($n = 1$ per sampling event) were stored in the dark at −80 °C. Water column samples of 4 L were collected at 2, 10, 20, and 50 m depths relative to sea level. Bottom sea ice samples were collected in two 5 cm thick sections. Samples were extracted in 90% acetone overnight at 4 °C and subsequently analyzed on a fluorometer (TD-700; Turner Designs, USA), including an acidification step (1 M HCl) to determine phaeopigments[51]. One subsample per underway sampling event was measured on board within 3 days after sampling, while a second subsample per underway sampling as well as the weekly samples were analyzed at the Alfred Wegener Institute (AWI) after the campaign had ended (5–9 months after sampling). No systematic difference between the replicates could be detected, indicating that no significant degradation of Chl-a took place during storage and transport. For sampling events with more than one replicate (i.e., the daily underway sampling), daily averages were used for further analyses (change point, rate of change).

## Particulate organic carbon measurements

Samples for measurements of POC concentrations were collected by filtering water on pre-combusted (450 °C, 4 h) glass fiber filters (GF/F; 0.7 μm nominal pore size) using mild vacuum. Filters ($n = 1$ per sampling event) were freeze-dried and carbonates were removed by contact with HCl fumes for 4 h in a vacuum-enclosed system. Filters were packed into tin cups and analyzed with an elemental analyzer (Flash 2000, Thermo Scientific, Milan, Italy) at the University of La Rochelle, France (Littoral, Environment and Societies Joint Research Unit stable isotope facility).

## Light microscopic analysis

Samples for light microscopic analysis ($n = 1$ per sampling event) were fixed with hexamethylenetetramine-buffered formalin solution (1% final concentration). Samples were stored at 4 °C in the dark until analysis. Samples from the water column were analyzed via inverse light microscopy at AWI (Germany), while samples from ice cores were analyzed at IOPAN (Poland).

## Flow cytometry

Samples for flow cytometric abundances, collected in triplicate, were preserved by adding 38 μL of glutaraldehyde to 1.5 mL of sample. Samples were kept for -2 h at 4 °C in the dark, flash-frozen in liquid nitrogen, and then stored at −80 °C until analysis. The abundance of pico- and nano-sized phytoplankton (approximate cell size 0.2–2 μm and 2–20 μm, respectively) were determined using an Attune® NxT, Acoustic Focusing Cytometer (Invitrogen by Thermo Fisher Scientific) with a syringe-based fluidic system and a 20 mW 488 nm (blue) laser. Cells were counted after thawing the sample ($n = 1$ per sampling event) and the various groups discriminated based on their red fluorescence (BL3) vs. side scatter (SSC), red fluorescence (BL3) vs. orange fluorescence (BL2) and orange fluorescence (BL2) vs. side scatter (SSC). The gating strategy is illustrated in Fig. S12.

## Primary production measurements

Potential NPP was determined by duplicate incubation ($n = 2$ per sampling event) with $NaH^{14}CO_3$ spike (53.1 mCi mmol$^{-1}$; Perkin Elmer; applied specific activity of 0.5 μCi mL$^{-1}$) for 24 h under reference conditions (1.0 ± 0.5 °C and 10 ± 3 μmol photons m$^{-2}$ s$^{-1}$) together with a dark control. For water column measurements, incubations contained 500 mL per sample, while sea ice incubation had volumes of 250–500 mL. After 24 h, incubated samples were filtered onto GF/F filters, acidified with 200 μL of 1 M HCl, and left to degas overnight. After addition of 10 mL of scintillation cocktail (Ultima Gold AB, PerkinElmer), samples were vortexed and left to stand in the dark for 20–30 h before counting on the liquid scintillation counter, using automatic quench correction and a counting time of 5 min. For blank determination, one replicate was immediately filtered and acidified. Per incubation bottle, 100 μL aliquots were mixed with 100 μL ethanolamine immediately after spiking to determine the total amount of added $NaH^{14}CO_3$. Subtracted blank values were on average 25% ($n = 23$) of the incubated sample counts. Potential NPP was calculated as

$$NPP = ([DIC]\left(DPM_{sample} - DPM_{0\%}\right)1.05)/(DPM_{100\%}t) \tag{1}$$

where [DIC] denotes the concentrations of dissolved inorganic carbon in the sample. $DPM_{sample}$ denotes the disintegrations per minute (DPM) in the samples, $DPM_{0\%}$ reflects the blank value, $DPM_{100\%}$ denotes the DPM of the total amount of $NaH^{14}CO_3$ added to the samples, and t is the duration of the incubation.

For the calculations, DIC data from the MOSAiC expedition[52] was used. The value of 1.05 is used to correct for fractionation against $^{14}$C relative to $^{12}$C[53]. C fixation in dark controls was subtracted to derive NPP.

## Data analysis

To detect the timing of photosynthetic biomass buildup initiation, we calculated the rate of change for daily measurements of underway Chl-a, using a 5-day interval[54], and detected the initiation as the first period in which dx/dt was positive for 5 days or more. Changepoint analysis[19] for daily underway Chl-a measurements were performed via the EnvCrp package in R[55]. Out of the twelve models in the package, six returned successful fits (the others failed as they would have required a larger dataset size). The six functional models returned three consecutive values (with corresponding dates) as the change points of the dataset. More details are provided in the supplementary information (Supplementary Note 1). The average of these six models was used with one standard deviation.

## Light measurements in the water column

Light measurements in the mixed water layer underneath the ice were obtained from OptiCALs[22], which are autonomous instruments that provide vertically resolved downwelling irradiance as a proxy for downward planar irradiance. Three OptiCALs were installed at the MOSAiC site, referred to as "hh", "gg", and "ee". OptiCAL hh was deployed at the SYI dark coring site (LM-site), OptiCAL ee at the so-called Ridge site, and OptiCAL gg at the so-called Monster site (Fig. S2). All three instruments were calibrated to absolute PAR intensity (photosynthetic photon flux density between 400 nm to 700 nm) with a remaining uncertainty of ±20%[56].

Because of inhomogeneities of the ice and snow, the raw measurements of the three OptiCALs have to be processed to obtain a robust average light field underneath the ice. In doing so, we took care to rather err towards an estimate of too much light, such that our resulting light intensity provides an upper bound of the true light intensity.

Our processing comprised two steps, with the first accounting for local inhomogeneities at the measuring sites, and the second for inhomogeneities across the ice cover. In the first step, we correct the individual light profiles at the measuring sites. These profiles generally follow an exponential decay with depth, as would be expected for a homogenous light field at the top of the water column and constant light-transmission properties of the water column[57]. However, since both these conditions are not perfectly fulfilled, slight deviations from an exponential decay curve were apparent in the profiles. For example,

the average light intensity across the narrow surface area sampled by the uppermost sensor can differ from the average light intensity across the much wider surface area sampled by the sensors further down. To correct for these local surface inhomogeneities we used the correction model of Laney et al[28]. The model eliminates near-surface shaded areas in the irradiance profiles. Occasionally, the model was unable to reliably approximate the measured light profile, resulting in an excessively high standard deviation of the fit (i.e., $\sigma > 1\,\mu mol$ photons $m^{-2}\,s^{-1}$) or an unreasonably high diffuse attenuation coefficient for such low-biomass waters (i.e., $\kappa > 0.06\,m^{-1}$ [58]); the affected profiles were left uncorrected (see Fig. S13 for example of a valid and an invalid fit). For the two lowest sensors at 32 m and 50 m depth, we found that a linear decay curve better represented the measured profiles, and therefore used a linear fit for this lowermost part of the profiles.

Having thus obtained profiles that consider the local inhomogeneities at the measurement site, in a second step we corrected for horizontal inhomogeneities in the light field underneath the entire ice cover. These horizontal inhomogeneities are almost exclusively related to differences in the snow thickness, as described in the main paper. The spatio-temporal variability of the snow cover across the MOSAiC site was obtained from snow-thickness transect data[59]. These transects showed that the snow cover at the MOSAiC site varied little between sea ice of different ages and remained largely unchanged between March and the melt onset in May. Spatial differences in snow thickness were mainly related to the three observed sea-ice surface types level ice, rubble ice, and deformed ice[31] of which the ice cover consisted to roughly equal shares. All optical sensors were deployed on level ice, which generally had the thinnest snow cover and the thinnest ice thickness, and therefore the highest light transmissivity. Snow thickness at the optical measurement sites was 0.19–0.21 m throughout March and April as measured by snow buoy 2019S96[60], which was deployed at the LM-site along with OptiCAL hh. Snow-thickness measurements at OptiCALs gg and ee gave very similar values. These snow thicknesses of around 20 cm were consistently lower than the mean snow thickness along transect lines S-loop and N-loop[59] which varied between 0.25 m and 0.30 m. This in isolation suggests that our light measurements are at the upper end of the true light intensity underneath the ice, as it was on average covered by thicker snow than at the measuring sites. However, this is countered by very high light transmission in regions with particularly thin snow thickness as found along the snow-thickness transect. To account for these inhomogeneities, we determined the ratio between the PAR-transmissivity ($T_{snow,PAR}$) averaged across the snow transects and the PAR-transmissivity averaged across the deployment sites. To obtain $T_{snow,\,PAR}$, we use a standard radiative transfer model for ice and snow[61] given as

$$T_{snow,PAR} = I_0 \cdot e^{-\kappa_{snow}(h_{snow} - 0.03\mathrm{m})} \qquad (2)$$

This model accounts for a surface scattering layer with an initial transmissivity of $I_0$ of the uppermost 3 cm of the snow cover and exponential attenuation below. The surface transmission value $I_0$ is irrelevant for our purposes, as its value cancels when calculating the ratio between different values of $T_{snow,\,PAR}$. The attenuation coefficient of snow for downwelling PAR irradiance $\kappa_{snow}$ was obtained from exponential fits to MOSAiC irradiance measurements above the snow[62] and below the snow[63]. These measurements result in a mean attenuation coefficient of $\kappa_{snow} = 14.1\,m^{-1} \pm 1.4\,m^{-1}$ for the snow layer at the MOSAiC site in March/April, including all diurnal solar zenith angles and sky conditions. The average value of $T_{snow,\,PAR}$ across all snow thickness measurements along the transect lines S-loop and N-loop from March and April[59], was 33 ± 13% higher than the average value of $T_{snow,\,PAR}$ at the deployment sites of the OptiCALs, primarily because of the high transmissivity in areas of exceptionally thin snow. To account

for the related potential underestimation of area-averaged transmitted light, we therefore corrected the raw measurements at the deployment sites by adding 33% to their values. Note that recent more advanced formulations of radiative transfer parametrizations[64] yield smaller biases of the order of 10% based on the MOSAiC measurements such that the light intensity after a correction of +33% provides an upper limit of the under-ice PAR.

The mean irradiance in the surface mixed layer was then finally calculated by spatially (3–50 m water depth) and temporally (24 h before ECO sampling) averaging the corrected light profiles. On 28 March, as conservatively identified as the day of onset of photo-synthesis, the three measurement sites gave average light intensities across the mixed layer of 0.014, 0.033, and 0.061 $\mu mol$ photons $m^{-2}\,s^{-1}$, giving an upper bound for the light intensity required for photo-synthesis of 0.04 ± 0.02 $\mu mol$ photons $m^{-2}\,s^{-1}$.

### Light measurements in the ice
Light measurements in the ice were obtained from a lightharp[65], an instrument that measures profiles of upward and downward irradiance in the ice. Calibration of the light harp to absolute PAR intensity was possible with a remaining upper bound for potential biases of 12.2%[65]. For this study, the calibrated light harp data were used as independent data to qualitatively examine the long-term stability and consistency of the OptiCAL data, to determine PAR directly at the underside of the ice, and to obtain the diffuse attenuation coefficient of snow from measurements directly at the snow−ice interface.

### Reporting summary
Further information on research design is available in the Nature Portfolio Reporting Summary linked to this article.

## Data availability
The data used in this study have been deposited in the PANGAEA repository (https://www.pangaea.de/). Cruise track data used in Fig. S1 is available as Kanzow (2020[66], https://doi.org/10.1594/PANGAEA.924681). Continuous light data from OptiCALs are available as Anderson et al., (2023a[67], https://doi.org/10.1594/PANGAEA.928495), Anderson et al., (2023b[57], https://doi.org/10.1594/PANGAEA.955045), and Anderson et al., (2023c[68], https://doi.org/10.1594/PANGAEA.954849). Continuous light data from the lightharp are available as Fuchs et al., (2024a,b[63,65], https://doi.org/10.1594/PANGAEA.963743). Chl-a concentration data from the underway system are available as Hoppe et al., (2023a[69], https://doi.org/10.1594/PANGAEA.963277) and the CTD rosette casts are available as Hoppe et al., (2023b[70], https://doi.org/10.1594/PANGAEA.962597). Flow cytometric cell count data from the water column are available as Müller et al., (2023a[71], https://doi.org/10.1594/PANGAEA.963430), and data from sea ice are available as Müller et al., (2023b[72], https://doi.org/10.1594/PANGAEA.963560). Data on microscopic cell counts are available as Kraberg (2024[73], https://doi.org/10.1594/PANGAEA.965913); the summarized data as used in this study are provided in the Supplementary Information (Table S1). Source data are provided with this paper.

## Code availability
The Python code to process PAR data in the described way is available under Fuchs (2024[74], https://doi.org/10.5281/zenodo.12772364). The R package EnvCPT (v1.1.3) used for change point analyses is available on GitHub (https://github.com/rkillick/EnvCpt/).

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

## Acknowledgements

Data used in this manuscript was produced as part of the international Multidisciplinary drifting Observatory for the Study of the Arctic Climate (MOSAiC) with the tag MOSAiC20192020 and the Project ID AWI_PS122_00. C.J.M.H., N.F., and D.N. received funding from the German Ministry of Education and Research (BMBF) through the project NiceLABpro (grant no. 03F0867A). D.N. received funding from the Deutsche Forschungsgemeinschaft under Germany's Excellence Strategy (EXC 2037; CLICCS—Climate, Climatic Change, and Society; grant no. 390683824). A.L., E.L., G.B., O.M., and P.A. and were supported by the Research Council of Norway through project HAVOC (grant no. 280292). BL was supported by a Fellowship at the Hanse-Wissenschaftskolleg Institute for Advanced Study, Delmenhorst, Germany. L.O. received funding from the BMBF-funded project nuArctic (grants 03F0918A). A.T. was supported by the Swedish Research Council VR (grant no. 2018-04685), the Swedish Research Council Formas (grant no. 2018-00509), and the Swedish Polar Research Secretariat (grant no. 2019-153), granted to Professor P. Snoeijs-Leijonmalm, Stockholm University, Sweden. We thank the cruise participants, ship's crew, and logistics support for MOSAiC legs 2 and 3 as well as everyone else who contributed to the realization of MOSAiC (Nixdorf et al. 2021). Particularly, we would like to thank those colleagues contributing to FYI coring and water sampling, i.e., A. Ulfsbo, S. Fons, S. Spahic, C. Marsay, L. Eggers, J. Grosse, G. Castellani, D. V. Divine, L. M. Olsen, S. Torres-Valdez, E. Damm, A. Dumitrascu, and P. Simões Pereira. We thank the MOSAiC leg 2 and 3 CTD teams, particularly J. Schaffer, for their help with water sampling. Sea ice light microscopy data was provided by M. Różańska-Pluta, A. Tatarek, and J. M. Wiktor. We thank L. Heitmann, T. Brenneis, and A. Terbrüggen for help with Chl-a

measurements. D.V. Divine, C. Katlein, P. Itkin, and I. Raphael are acknowledged for deploying the OptiCALs and the lightharp. We thank G. Birnbaum for the coordination and recording of aerial images. Y. Novak is acknowledged for producing the schematic figure of the sampling sites. Invaluable support, planning, and coordination were provided by A.A. Fong and R. Gradinger.

## Author contributions

C.J.M.H. developed the research question and concept. C.J.M.H., D.N., and N.F. developed the overall storyline. C.J.M.H. and A.T. conducted field measurements. C.J.M.H., A.K., and O.M. measured biological samples. CJMH analyzed biological data. N.F. analyzed the optical measurements. CJMH and DN wrote the paper, with contributions from N.F., P.An., P.As., J.B., G.B., G.G., A.K., A.L., B.L., E.L., M.L., O.M., L.O., B.R., B.S., A.T., and J.W.

## Funding

## Competing interests

The Authors declare no competing interests.
