## [Peer Review File · Nature Communications]

Photosynthetic light requirement near the theoretical minimum detected in Arctic microalgaeREVIEWER COMMENTS

Reviewer #1 (Remarks to the Author):

GENERAL COMMENTS:

Overall, this is a well-written manuscript and provides significant results to the field. The manuscript contributes to filling the gap of the lower bound of light level sustaining photosynthesis and net primary production in the global ocean, which is still unknown, using the data from the MOSAiC field campaign in the central Arctic Ocean. This new lower limit (deeper euphotic zone) could reduce the uncertainty of the estimate of the net primary production of the world's ocean.

My major concerns, however, are:

- 1) I disagree with the claims in Lines 65-67 that "the constant ratios of carbon fixation to Chl-a (Figure S3) and particular organic carbon (POC) to Chl-a (Figure S4) throughout the study period". It is too strong. In particular, in Figure S4, where I found the ratio of POC:Chl-a increased from $\sim 1200 \text{ g g}^{-1}$ to $\sim 2000 \text{ g g}^{-1}$ ($\sim 66\%$). This may suggest the considerable role of photophysiological processes, which must be discussed. I agree with the wording "the lack of trend" used in the caption. In addition, the Mann-Kendall Trend test or similar may also strengthen the statistical test of "the lack of trend".
- 2) The current presentation of the manuscript assumes that the reader knows about the MOSAiC expedition or has to read many MOSAiC-related papers before fully understanding these papers. "Figure 0", which horizontally contains more information on site locations of RV Polarstern and sensors (Figure 2) including the opened lead on April 17. and vertically shows the key depths within and underneath the sea ice (e.g., 11 m, 20 m, 50 m) mentioned throughout the manuscript, will improve the communication of the manuscript.
- 3) Adding snow depth and ice thickness data and discussion. The discussion at L148-153 fell short without snow depth range. I agree that the differentiation between first-year and second-year ice is not needed for the purpose of this study. However, adding the snow depth and ice thickness evolution as Figure 1C could help strengthen the robustness of the manuscript.

SPECIFIC COMMENTS:

Note that I do not comment on language issues such as spelling or grammatical mistakes.

- To avoid possible confusion, I suggest using "Arctic marine microalgae" in the title.
- Is "J. Wolka" a full name? If not, please edit to match with other authors in the list.
- L117 r2 but L334 R2. Please be consistent.
- The snow depth range should be reported, not only compared at L142-144.
- Increase Fig 1's marker size to be similar to Figure 3. Use different symbols, not only colour, to distinguish different reference conditions.
- Fig 1B 20m? or 11m? as the caption mentions. Please clarify.

Key results.

- L320 Redefine Chlorophyll-a for figure's caption.
- Add the uncertainty discussed at L502-504 as shaded patches in Figure 2.
- L424: "9 cm inner diameter"
- L438: 2 m depth relative to ice/ocean interface or sea level?
- L444 ... Institue (AWI)

Reviewer #2 (Remarks to the Author):

This paper, based on data from a recent Arctic cruise, is very interesting and worth publishing. The methods are sound and the data are reliable. The particular conclusion, that the authors have detected and confirmed primary production in the field at a light intensity of just $0.04 \mu\text{mol photons m}^{-2} \text{ s}^{-1}$ photosynthetically active radiation (PAR), is challenging and of great interest.

In my opinion, the greatest issue with this is whether the methods for measuring PAR in this study allow that conclusion. The measurements are detailed and the carefully performed, using state of the art equipment, but the value of $0.04 \mu\text{mol photons m}^{-2} \text{ s}^{-1}$ in Fig. 3 is (necessarily) based on an extrapolation. I do not think this allows such a strong conclusion that productivity occurred at this PAR. The paper is definitely worth publishing, but the conclusions need to be toned down. This is particularly in view of the recent discoveries in studies such as your reference (17) (Kennedy et al. 2019), with substantial dark fixation processes occurring at very low PAR values.

Some minor issues:

- The sentence from lines 134-136 is a bit clumsy and needs rewriting.

- Referring to 14C incubation as 'potential' net primary production is not useful, as recent studies continue to identify that this method can produce rates that vary anywhere between gross and net photosynthesis, depending on length of incubation and other factors. Just call it 14C productivity.

Reviewer #3 (Remarks to the Author):

The paper is a potentially very interesting study that may push the measurements of the known experimentally measured lower limit for marine photosynthesis an order of magnitude lower towards a proposed lower theoretical limit. The work from the previous measurement (reference 1 of the MS) of the lower limit was greeted with much excitement by the sea ice community, biologists more widely, and workers in the fields of Astro biology. Nature communications maybe the right communication channel for such an exciting claim. Much of the measurements, whilst extremely comprehensive and large in scale owing to the size of the MOSAIC campaign are standard and whilst they are very important data to have, the noteworthy part of the study is the pushing of the experimental limit an order of magnitude lower, and perhaps almost reaching the proposed limit. If I may voice an opinion, I would suggest that is very exciting.

There are two aspects to this work that are of equal importance. A series of biological experiments which are well explained, well documented with plenty of evidence presented and are of the standard in the field, and a series of physical light measurements that are not well explained, poorly evidenced and include some rather mysterious factors of ~ 3 or $1/3$ that are not well justified with data or reasoning as to why they are needed. The latter should be shown some more consideration.

Specific issues:-

1) The positioning of the light sensors is unclear relative to the ice/snow surface

2) The positioning of the biological samples, relative the position of the light measurements is unclear. There is likely to be great heterogeneity in the PAR light field under the ice in the

plane of the ice surface and it is not clear that the light field measurements are collated with the biological measurements which would be troubling.

3) It would be impossible from the description given to estimate or calculate or even judge as reasonable the intensity of light penetrating the snow and sea ice as there is no record of snow depth and type, ice depth and type.

4) Why does a correction of a third need to be made to the PAR measurement. There is not enough detail in the paper to reproduce this calculation and after reading several times I cannot see why this was done well enough to explain to another colleague.

5) If the PAR data need correcting by a factor that is "estimated" to be a third is the order of magnitude claim change in threshold safe to plus or minus 50%?

6) Beer-Lambert law decay of light in snow is very, depend on SZA unless the illumination source is diffuse and an albedo of 0.7 needs to be justified.

7) The averaging of PAR measured by the three PAR sensor array moorings vertically and horizontally: Do the authors mean a true average? Averaging data can lead to troubles, and I think some depiction of the data in supplementary to demonstrate there are no extremes, vertically or horizontally. Also is there any evidence that this 50m layer is well mixed? It seems a thick layer to average over. This reviewer did start downloading the PAR data (the authors should be congratulated for making it all available) to check these claims, but came to the conclusion that this was really something the authors should present in their supplementary information.

Thus unfortunately this reviewer is of the opinion that they would struggle to reproduce the conclusions of the physical light measurements from the level of the detail provided. The reviewer wishes to stress that this is a plea for revision and more explanation and evidence presented, not a rejection, as it is potentially exciting.

REVIEWER COMMENTS & AUTHOR RESPONSES

Reviewer #1 (Remarks to the Author):

GENERAL COMMENTS:

Overall, this is a well-written manuscript and provides significant results to the field. The manuscript contributes to filling the gap of the lower bound of light level sustaining photosynthesis and net primary production in the global ocean, which is still unknown, using the data from the MOSAiC field campaign in the central Arctic Ocean. This new lower limit (deeper euphotic zone) could reduce the uncertainty of the estimate of the net primary production of the world's ocean.

My major concerns, however, are:

1) I disagree with the claims in Lines 65-67 that "the constant ratios of carbon fixation to Chl-a (Figure S3) and particulate organic carbon (POC) to Chl-a (Figure S4) throughout the study period". It is too strong. In particular, in Figure S4, where I found the ratio of POC:Chl-a increased from ~ 1200 g g⁻¹ to ~ 2000 g g⁻¹ (~66%). This may suggest the considerable role of photophysiological processes, which must be discussed. I agree with the wording "the lack of trend" used in the caption. In addition, the Mann-Kendall Trend test or similar may also strengthen the statistical test of "the lack of trend".

We thank the reviewer for pointing out this incorrect wording, and we agree that these ratios should not be called 'constant' given the significant variability in the data stemming from particularly low Chl-a and POC concentrations. Following the reviewer's suggestion we now instead refer to a "lack of trend" in the main text (L69-71 in track changes version) as well as in the figure caption (Figure S5), and support this statement by results from the Mann-Kendall Trend test ($S=-1$, p value = 0.5). Given this lack of trend, and in combination with other data such as Chl-a specific carbon fixation under reference conditions (Figure S4), our statement regarding the lack of physiological acclimation nevertheless remains correct.

2) The current presentation of the manuscript assumes that the reader knows about the MOSAiC expedition or has to read many MOSAiC-related papers before fully understanding these papers. "Figure 0", which horizontally contains more information on site locations of RV Polarstern and sensors (Figure 2) including the opened lead on April 17. and vertically shows the key depths within and underneath the sea ice (e.g., 11 m, 20 m, 50 m) mentioned throughout the manuscript, will improve the communication of the manuscript.

We thank the reviewer for this useful suggestion. In response, we have added a conceptual figure illustrating the general setup and the different sampling methods and depths in our new Figure 1. As it proved difficult to include all relevant aspects into one figure, we also provide an additional figure in the supplement (Figure S2) showing the specific locations of different sampling points in the MOSAiC study area. We also would like to emphasize that the specific distances of sampling locations on the ice surface are not relevant for this study, as the sea ice moved about six times faster along the transpolar drift compared to the underlying water column (Schulz et al. 2023 preprint). This information has been added to the manuscript (L353-355, L569-571).

3) Adding snow depth and ice thickness data and discussion. The discussion at L148-153 fell short without snow depth range. I agree that the differentiation between first-year and second-year ice is not needed for the purpose of this study. However, adding the snow depth and ice thickness evolution as Figure 1C could help strengthen the robustness of the manuscript.

Thanks for pointing this out. We have now added additional information to the text and linked the text more closely to the comprehensive study of Itkin et al. 2023, in which the authors provide a detailed overview of the snow and ice evolution on MOSAiC. We now specify that

"The snow cover at the MOSAiC site spatially showed little difference between ice ages and temporally varied little between February and the onset of melt in May" (L161-163) Because of this existing study, we have decided not to reproduce the information in one of our figures.

SPECIFIC COMMENTS:

Note that I do not comment on language issues such as spelling or grammatical mistakes.

- To avoid possible confusion, I suggest using "Arctic marine microalgae" in the title.

Given we present data on marine and ice-associated microalgae, we feel that this suggestion would be imprecise, so we prefer to keep the title as before. Please note that the abstract specifies that data originates from the central Arctic Ocean.

- Is "J. Wolka" a full name? If not, please edit to match with other authors in the list.

Thanks for pointing this out, which has been corrected in the manuscript and the supplement.

- L117 r2 but L334 R2. Please be consistent.

Following the reviewer's comment, we now consistently use 'r2' throughout the manuscript (L124, L375, L376).

- The snow depth range should be reported, not only compared at L142-144.

Since we are focusing on the general statement that ice age was not a significant discriminator for light transmissivity, and we were less interested in the exact dependence on different snow thicknesses, we have decided to keep this section very general. The previously mentioned study by Itkin et al. (2023) provides a comprehensive overview of snow and ice conditions on MOSAiC. The authors showed that the snow cover on different ice ages harmonized in February. We added that point to the text as it provides important additional information and guides interested readers to the publication of Itkin et al. (L585-598). We hope this represents a good compromise for the reviewer between the necessary information and the length of the manuscript.

- Increase Fig 1's marker size to be similar to Figure 3. Use different symbols, not only colour, to distinguish different reference conditions.

Following the reviewer's suggestion, marker and font size have been harmonized between figures and different symbols have been used to distinguish datasets within subplots (new Figure 2).

- Fig 1B 20m? or 11m? as the caption mentions. Please clarify. Key results.

Thanks for spotting this error, the correct value is 11m and has been corrected in the axis title (L358-361).

- L320 Redefine Chlorophyll-a for figure's caption.

Done (L360, L373).

- Add the uncertainty discussed at L502-504 as shaded patches in Figure 2.

Done (L363-368).

- L424: "9 cm inner diameter"

Done (L485).

- L438: 2 m depth relative to ice/ocean interface or sea level?

We refer to 2 m relative to sea level, and have added this information to the text (L500-501).

- L444 ... Institutue (AWI)

'AWI' has been added (L506).

Reviewer #2 (Remarks to the Author):

This paper, based on data from a recent Arctic cruise, is very interesting and worth publishing. The methods are sound and the data are reliable. The particular conclusion, that the authors have detected and confirmed primary production in the field at a light intensity of just 0.04 $\mu\text{mol photons m}^{-2} \text{ s}^{-1}$ photosynthetically active radiation (PAR), is challenging and of great interest.

In my opinion, the greatest issue with this is whether the methods for measuring PAR in this study allow that conclusion. The measurements are detailed and the carefully performed, using state of the art equipment, but the value of 0.04 $\mu\text{mol photons m}^{-2} \text{ s}^{-1}$ in Fig. 3 is (necessarily) based on an extrapolation. I do not think this allows such a strong conclusion that productivity occurred at this PAR. The paper is definitely worth publishing, but the conclusions need to be toned down. This is particularly in view of the recent discoveries in studies such as your reference (17) (Kennedy et al. 2019), with substantial dark fixation processes occurring at very low PAR values.

We thank the reviewer for their interest in this manuscript, as well as for their useful comments. Regarding our light measurements, we now provide much more detail in the methods and make more explicit that the derived light levels are a conservative upper bound of the actual light field, with the true light levels being potentially even lower (e.g. L151-153, L171). Nevertheless, we also toned down our conclusions in some instances (e.g. L209-214, L218-220).

Regarding the uptake of organic carbon as an energy source under low and no light conditions, we agree that this probably plays an important role in winter survival. However, as we now write in the manuscript "Given that alternative energy sources such as respiration of storage compounds or mixotrophic uptake of organic matter have been available throughout winter and are not related to the return of sun light^{18,34,35}, the exponential increase in Ch-a concentrations must be a direct response to the measured increase in PAR" (L172-175).

Some minor issues:

- The sentence from lines 134-136 is a bit clumsy and needs rewriting.

Following the reviewer's suggestion, the sentence has been rewritten. It now reads "To determine the mean irradiance that primary producers in the water column were exposed to, we averaged the light fields horizontally across these three sites that continuously drifted with the ice over the underlying ocean^{23,24}" (L142-144).

- Referring to 14C incubation as 'potential' net primary production is not useful, as recent studies continue to identify that this method can produce rates that vary anywhere between gross and net photosynthesis, depending on length of incubation and other factors. Just call it 14C productivity.

Following the reviewer's comment, we replaced 'Potential net primary production' by 'potential ¹⁴C productivity' in all relevant figure axis labels. We would like to keep the term 'potential', however, as it indicates that ¹⁴C productivity has been measured under lab conditions with a fixed light level rather than under in situ conditions with increasing

irradiances. In the text, we now also use the term ^{14}C productivity, and then introduce NPP as a simplification aiming at better readability for the remainder of the text (L60-63). Further, we now clarify that our measurements were conducted over 24h (L60-63), thus being closer at net rates than shorter incubations would be.

Reviewer #3 (Remarks to the Author):

The paper is a potentially very interesting study that may push the measurements of the known experimentally measured lower limit for marine photosynthesis an order of magnitude lower towards a proposed lower theoretical limit. The work from the previous measurement (reference 1 of the MS) of the lower limit was greeted with much excitement by the sea ice community, biologists more widely, and workers in the fields of Astro biology. Nature communications maybe the right communication channel for such an exciting claim. Much of the measurements, whilst extremely comprehensive and large in scale owing to the size of the MOSAIC campaign are standard and whilst they are very important data to have, the noteworthy part of the study is the pushing of the experimental limit an order of magnitude lower, and perhaps almost reaching the proposed limit. If I may voice an opinion, I would suggest that is very exciting.

There are two aspects to this work that are of equal importance. A series of biological experiments which are well explained, well documented with plenty of evidence presented and are of the standard in the field, and a series of physical light measurements that are not well explained, poorly evidenced and include some rather mysterious factors of ~ 3 or $1/3$ that are not well justified with data or reasoning as to why they are needed. The latter should be shown some more consideration.

We are very happy that the reviewer finds our paper exciting and worth publishing in Nature Communications. To address the reviewer's concern regarding our description of the light measurements, these are now described in much greater detail, in particular regarding our approach to correct the data in order to obtain an upper bound on the true light field (see below).

Specific issues:

- 1) The positioning of the light sensors is unclear relative to the ice/snow surface.

Following the reviewer's comment, we now specify that the depth levels of the light sensors relate to the ice surface (L139-142).

- 2) The positioning of the biological samples, relative the position of the light measurements is unclear. There is likely to be great heterogeneity in the PAR light field under the ice in the plane of the ice surface and it is not clear that the light field measurements are collated with the biological measurements which would be troubling.

To address this helpful comment, we now explain this in greater detail, both in the main text (L142-157, 353-355) and in the supplemental material (L570-572). As the ice and the ocean are moving at different speeds, the relative positions of the sensors and sampling locations are of minor importance for the interpretation of our data. Due to this circumstance, we have calculated an upper bound of the potentially available light to the organisms in the upper mixed layer.

3) It would be impossible from the description given to estimate or calculate or even judge as reasonable the intensity of light penetrating the snow and sea ice as there is no record of snow depth and type, ice depth and type.

The study focuses on the light sensitivity of the under-ice microalgae. Even though we certainly share the enthusiasm for researching light transmissivity, we had to keep this aspect very brief in order to keep the manuscript short. Hence, we specifically address transmissivity only in the consideration of possible underestimation of PAR due to large-scale differences in snow thickness. Based on the reviewer's feedback, we have nevertheless expanded the related discussion in the methods section and have strengthened the link to the comprehensive study of snow and ice conditions on MOSAiC by Itkin et al. (2023).

4) Why does a correction of a third need to be made to the PAR measurement. There is not enough detail in the paper to reproduce this calculation and after reading several times I cannot see why this was done well enough to explain to another colleague.

We would like to thank you for your feedback, which has motivated us to reformulate the entire section and provide a better overview of the procedure in the main text (L136-157) and the methods section (L578-625). In particular, we now make it more explicit that all our corrections increase the estimated light intensity, such that our results are an upper bound for the true light intensity. We realized from the reviewer's comments that this motivation was insufficiently explained.

5) If the PAR data need correcting by a factor that is “estimated” to be a third is the order of magnitude claim change in threshold safe to plus or minus 50%?

See above: All corrections aim at increasing light intensity, such that our overall result is an upper bound on the true light intensity.

6) Beer-Lambert law decay of light in snow is very, depend on SZA unless the illumination source is diffuse and an albedo of 0.7 needs to be justified.

We fully share this assessment of using the Beer-Lambert law to describe the decay of light in snow in the case of detailed studies of light transmissivity. However, since we are dealing here with a general offset factor for in situ under-ice measurements, and aim for estimating the upper bound of the light intensity as described above, we decided to average over all the different conditions and obtained an attenuation coefficient that fits well into the range of previously published values.

We now better explain the derivation of the values, the use of the surface layer reflectivity, and the general scheme. In addition, we have improved the error estimation of the determined factor by exploring the impact of deviations within the range of the standard deviation of the attenuation coefficient.

Recently, an exciting study was published (Roche and King, 2024) that provides an improvement of the exponential scheme. We have tested this and obtained a smaller correction factor than was employed based on our scheme, which further strengthens the credibility of our upper limits. Roche and King furthermore analyzed the impact of the SZA and derived a correction coefficient for different angles. This SZA correction and the surface layer reflections (in the manuscript now: surface transmission value) are linear factors in the transmissivity formulation and thus are canceling out in the comparison between area-averaged transmissivity and local transmissivity at the deployment sites. We, therefore, did not investigate their standard error further.

7) The averaging of PAR measured by the three PAR sensor array moorings vertically and horizontally: Do the authors mean a true average? Averaging data can lead to troubles, and I think some depiction of the data in supplementary to demonstrate there are no extremes, vertically or horizontally. Also is there any evidence that this 50m layer is well mixed? It seems a thick layer to average over. This reviewer did start downloading the PAR data (the authors should be congratulated for making it all available) to check these claims, but came to the conclusion that this was really something the authors should present in their supplementary information.

As part of the revision of the light data processing description in the methods parts, we added two example profiles of the under-ice light measurements (Figure S12). One shows a typical profile with near-surface shading, that was successfully corrected with the Laney model. The other shows a typical profile that could not be corrected, due to an invalid fit. However, these profiles themselves did not actually look wrong, but due to some minor variability, they were not suitable for the fit. In consequence, they were left raw. Only in late April, when leads opened during a strong deformation event profiles of OptiCAL gg were temporarily strongly disturbed (we added that note to the method description, too). Overall, however, the vast majority of profiles showed a smooth evolution with depth, and agreed well among the three measurement sites. We therefore trust that a simple averaging is sufficient for our purposes.

Regarding the mixed layer, all available data sources (e.g. Chl-a concentrations, Microstructure Profiler derived mixing rates and Brunt-Väisälä frequency distribution) suggest that the upper 50m of the water column were actively mixed during our study periods. This is now more clearly described and referenced in the methods section (L144-150).

Thus unfortunately this reviewer is of the opinion that they would struggle to reproduce the conclusions of the physical light measurements from the level of the detail provided. The reviewer wishes to stress that this is a plea for revision and more explanation and evidence presented, not a rejection, as it is potentially exciting.

We are glad that the reviewer considers our study as suitable for publication, and hope that the added detail and explanations in the main text and the methods section as well as the additional figures on the spatial aspects of the setup resolve this concern.

REVIEWERS' COMMENTS

Reviewer #1 (Remarks to the Author):

I feel that the authors have thoroughly addressed all my concerns raised during the initial review process.

The manuscript has been heavily improved in terms of content, structure, and presentation. The language quality remains very high, and the new Figure 1 enhances the clarity of the observations. The alignment of the manuscript's structure with the research goals and methods is now well-defined, making the study's contributions to the field more evident.

However, I recommend the authors:

- to revise the Figure 1 caption by including the full form of "Chl-a", "NPP", and "OptiCAL" (also in Figure 3 for OptiCAL) and

- to fix very minor comments:

L159-160 When reporting sea ice thickness, please use SI unit, i.e., 1.14 m instead of 114 cm.

L175 "Chl-a".

Thank you again to the authors for their diligence in documenting the changes made in response to the reviewers' comments.

RESPONSE TO REVIEWERS' COMMENTS

Reviewer #1 (Remarks to the Author):

I feel that the authors have thoroughly addressed all my concerns raised during the initial review process.

The manuscript has been heavily improved in terms of content, structure, and presentation. The language quality remains very high, and the new Figure 1 enhances the clarity of the observations. The alignment of the manuscript's structure with the research goals and methods is now well-defined, making the study's contributions to the field more evident.

We thank the reviewer for their constructive feedback and the kind words.

However, I recommend the authors:

- to revise the Figure 1 caption by including the full form of "Chl-a", "NPP", and "OptiCAL" (also in Figure 3 for OptiCAL) and

Done as suggested (L1348-1352, L1372-1373)

- to fix very minor comments:

L159-160 When reporting sea ice thickness, please use SI unit, i.e., 1.14 m instead of 114 cm.

Done as suggested (L185-186)

L175 "Chl-a".

Done as suggested (L201)

Thank you again to the authors for their diligence in documenting the changes made in response to the reviewers' comments.